# Mechanical Testing of Epoxy Resin Modified with Eco-Additives

**DOI:** 10.3390/ma16051854

**Published:** 2023-02-24

**Authors:** Agnieszka Derewonko, Wojciech Fabianowski, Jerzy Siczek

**Affiliations:** 1Faculty of Mechanical Engineering, Institute of Mechanics and Computational Engineering, Military University of Technology, Sylwestra Kaliskiego 2, 00-908 Warsaw, Poland; 2Military Institute of Chemistry and Radiation, gen. Antoniego Chruściela 105, 00-910 Warsaw, Poland

**Keywords:** epoxy resin, eco-additives, experimental tests

## Abstract

The future belongs to biodegradable epoxies. In order to improve epoxy biodegradability, it is crucial to select suitable organic additives. The additives should be selected so as to (maximally) accelerate the decomposition of crosslinked epoxies under normal environmental conditions. However, naturally, such rapid decomposition should not occur within the normal (expected) service life of a product. Consequently, it is desirable that the newly modified epoxy should exhibit at least some of the mechanical properties of the original material. Epoxies can be modified with different additives (such as inorganics with different water uptake, multiwalled carbon nanotubes, and thermoplastics) that can increase their mechanical strength but does not lead to their biodegradability. In this work, we present several mixtures of epoxy resins together with organic additives based on cellulose derivatives and modified soya oil. These additives are environmentally friendly and should increase the epoxy’s biodegradability on the one hand without deteriorating its mechanical properties on the other. This paper concentrates mainly on the question of the tensile strength of various mixtures. Herein, we present the results of uniaxial stretching tests for both modified and unmodified resin. Based on statistical analysis, two mixtures were selected for further studies, namely the investigation of durability properties.

## 1. Introduction

Composites are widely used in the automotive and aerospace industries. Their advantages include low weight and infinite durability. Unfortunately, infinite durability is also a disadvantage. A large amount of non-biodegradable waste is generated. Thus, a method is needed to recycle the waste. Pyrolysis is the most commonly used method of recycling in the case of carbon-fiber-reinforced composites with an epoxy matrix (CFRP) [1]. The recovered carbon fibers are used in thermoplastic and thermosetting coatings and nonwoven fabrics. On the other hand, the recycling of glass fiber reinforced polymers (GRP) is carried out via their simultaneous processing in the cement kilns with fuels from waste. Thermoplastic composites are also crushed and melted. Currently, popular methods for reducing environmental littering when using biodegradable or biocompostable polymers also have some disadvantages. One of them is the pollution of the aquatic environment by slowly degraded materials, mainly those that are hydrolyzed. For instance, in the case of epoxy–glass composites, there is a risk of releasing bisphenol A (BPA), which is highly toxic [2].

The degree of the composite degradation was determined using the methods utilized in the evaluation of the composite cracking resistance [3,4]. An effective and safe method for obtaining biodegradable or biocompostable composites also includes the issue of their mechanical strength in the assumed period of use (operation), which is often neglected. Therefore, the following question arises: How does an organic additive influence the mechanical strength of the epoxy resin?

Scientific and research literature on the use of organic additives is extensive and relates mainly to compounds modifying interactions at the interface between the glass (carbon) fiber and the liquid epoxy matrix [5,6,7]. Numerous additives—both organic and inorganic—were tested as epoxy-modifying agents [8,9,10]. Note that even small amounts of added compounds, less than 1% *w*/*w*, can change the mechanical properties of epoxy composites in a significant way. The structural integrity of the composites and the interface of bonded layers, such as metal and composite, is ensured by adequate adhesion. The role of such issues in relation to the mechanical properties of structures was described by Bellini and his team in a 2019 paper [11]. However, the issue of modifying matrix polymers for CFRP composites in terms of strength was discussed, among others, in works [12,13,14,15].

One type of additive used in epoxy–glass composites is epoxidized natural oils that change the properties of the epoxy matrix, mainly by increasing their mechanical strength and impact strength. The effect of adding inedible oils of natural origins, e.g., karanja oil from Pongam tree seeds, to epoxy resins is described in [16]. In the case of composites, a series of works was carried out on the study, assessment and modeling of the effect of homophasic and heterophasic additives on mechanical properties, impact resistance and thermal resistance, e.g., [17]; however, they do not take into account their biodegradability.

Particularly interesting are the works on adding organic substances, including epoxidized edible and inedible oils known and used as additives for diesel fuels, e.g., biodiesel [18]. These substances, as rapidly biodegradable, cannot be used in composites for which their expected aging resistance is counted in years. Therefore, using organic substances that are more durable than biodiesel additives is proposed, namely, epoxidized cellulose derivatives, methyl cellulose, carboxymethyl cellulose and similar substances [19]. In work [20], it was shown that the chemical modification of cellulose pulp by epoxidation reactions has a positive effect on the rheological properties of the final product.

Environmental pollution and climate protection have intensified work on ecological additives that limit the extraction of fossil raw materials and enable the recycling of products after their use. A novel degradable and recyclable thermoset hyperbranched epoxy resin (EFTH-n) synthesized from bio-based 2,5-furandicarboxylic acid was described in work [21]. It has been demonstrated that EFTH-n is successfully used to improve the toughness, strength, modulus and elongation of DGEBA (Bisphenol A diglycidyl ether).

The issue of biodegradability has been raised frequently in recent times. This is due not only to massive and growing environmental pollution but also dwindling fossil resources. Among the many works from 2020, those in which the chemical aspect is related to the strength and durability of materials, such as [22,23,24], are included. As early as 2021, in paper [25], fully recyclable epoxy formulations using organic waste flour have been proposed. In contrast, paper [26] proposed environmentally friendly adhesives for aerospace applications. A comprehensive list of current works in the field of the application of clove oil in the production of composites can be found in Matykiewicz and Skórczewska, 2022 [27].

Although a new type of hardener was patented in 2021, Recylamine, which makes the epoxy resin biodegradable after curing, is not the only hardener used in the pyrolysis process; the Z-1 (triethylene tetramine) hardener is still used. The study investigated the effect of various organic additives and their content on the tensile strength of rowing specimens made from them.

## 2. Materials and Methods

The main objective of the study is to determine the tensile strength of specimens made of epoxy resin mixtures with natural additives.

A change in the crosslinking of a polymer is achieved by introducing an additive in a certain proportion. After a set finite time, such a change should cause the failure of the cured epoxy resin. A gradual degradation of the polymer crosslinking will allow for the separation of the epoxy matrix from fibers, which leads to the partial degradation of the composite and the segregation of its components. The types of analyzed mixtures are listed in Table 1. In Table 1, organic additives are as follows: ESO—epoxidized soybean oil (Boryszew S.A.); MC—methyl cellulose (C.T.S.); EC—ethyl cellulose (C.T.S.); CMC—carboxymethyl cellulose (C.T.S.). In Table 1, phr means parts by weight per 100 parts by weight of the resin. Epidian 601 is the epoxy resin (Zakłady Chemiczne Ciech Sarzyna).

Some basic properties of Epidian 601 are collected in Table 2.

All mixtures were prepared using the method described below. In total, 100 g of Epoxy resin 601 was RT mixed up with 3 g or 10 g of organic additive, and 13 g of Z-1 crosslinking agent (triethylene tetramine from Zakłady Chemiczne Ciech Sarzyna) was added next, vigorously mixed for 5 min, and vacuum degassed for 10 min. All specimens of Epidian 601 with the organic additive and crosslinking agent were at first, right after addition, opaque and more viscous, but within 2–3 min after mixing, the specimens turned yellow again and were more transparent and less viscous, similarly to the starting Epidian 601 resin. Only after the addition of 3 phr or 10 phr of ethyl cellulose to the Epidian 601 resin did the specimens turn into an opaque white color and were more viscous; even after prolongated RT mixing for 10 min, they still remained unchanged. This observation suggests that EC, being more hydrophobic than other cellulose derivatives, was incompatible with Epidian 601 resin, forming a separate phase that weakly interacted with the surrounding resin. This was later confirmed by the poor mechanical properties of the Epidian/EC system. In preparation of Epidian/organic additive/crosslinking agent Z-1, it should be remembered that all mixing/degassing procedures must be completed within 30 min, because after 40 min, these systems start to gel, and no mixing or degassing is possible.

The described mixtures were used to create test specimens for uniaxial tensile tests. Polastosil AD4 (Zakłady Chemiczne Ciech Sarzyna) was used to prepare the silicone mold. There have to be holes in the mold that match the shape and dimensions of the specimens that will be created. The reference specimens shown in Figure 1a were used to create them. Both the mold and the reference specimens were made by 3D printing technology.

The silicone molds with holes are shown in Figure 1b. Test specimens for the uniaxial tensile test were formed from mixtures 1 to 9 in Table 1.

In Figure 2a, the reference structure of the epoxy resin taken with the use of a digital microscope Keyence VHX 6000 with a magnification of 200× is shown. The effect of organic additives on the structure of the modified epoxy resin is shown in Figure 2b–i. The scale on each picture allows the determination of the size and distribution of additives. Inserting a scale is more practical than using a scale in pictures.

## 3. Uniaxial Stretching Tests

Biodegradable epoxy resins should indicate physical and chemical properties, as well as mechanical properties, that are similar to unmodified resins with practically unlimited disintegration time. Therefore, experimental uniaxial stretching tests according to ASTM638 were conducted on test specimens made of epoxy resins with an organic additive. A series of reference test specimens made of unmodified epoxy were also examined. The nominal dimension of the test specimen is shown in Figure 3.

Proper experimental testing requires additional specimen preparation. The specimens were cleaned and specially marked. The places of grips have been marked, as well as measurement points (black dots) spaced 30 mm apart. An exemplary set of test specimens prepared for testing that are made of epoxy resin E601 with the addition of ESO10 (epoxidized soybean oil) is shown in Figure 4.

The KAPPA 50 DS electromechanical loading system with a ZwickRoell video extensometer (Figure 5a) and specialized software was used for the tests, enabling the simultaneous measurement of the elongation and change in the width of the specimen in a given area. The machine ensured precise axial alignment to ASTM E292. ZwickRoell videoXtens uses image processing, allowing longitudinal and transverse strains to be determined with greater accuracy.

The specialized testXpert ZwickRoell (Figure 5b) software recorded measurement data such as time, distance, force, elongation and width change in the measurement area. The specimens were stretched at a speed of 2 mm/min until they were damaged. The frequency of data acquisition was set at 10 Hz due to the static nature of the load. Obtained in tensile test stress–strain curves are shown in Figure 6.

## 4. Results and Discussion

The tensile test results were developed for each specimen separately. The tensile Young’s modulus E, the ratio of the transverse strain to longitudinal strain in the uniaxial stress state (Poisson’s ratio *ν*) and destructive stresses and strains, *σ_f_* and *ε_f_*, respectively, were determined for each group of specimens after rejecting the extreme results in the group. Young’s modulus E was determined on the basis of the slope of the diagram of stresses as a function of strains. Poisson’s ratio is a measure of deformation and has been defined as the slope angle of the transverse strain curve versus the longitudinal strain. The values of failure stresses and strains were determined as extreme stress and the corresponding strain of the stress function—longitudinal strain. The average values of Young’s modulus, Poisson’s ratio, stresses and failure strains together with the number of specimens are summarized in Table 3. The standard deviation (SD) values for each quantity are also included in Table 3.

The standard deviation, which is a measure of the width of the value scattering from the mean value, was determined using Excel according to(1)$$\sqrt{\frac{\sum \left(x-\bar{x}\right)}{\left(n-1\right)}},$$
where $\bar{x}$ is the sample mean, and *n* is the sample size.

Figure 7 shows the average values of Young’s modulus, with error bars showing the standard deviation values as a function of the specimen type.

Figure 8 places the average values of Poisson’s ratio as well as standard deviation bars as a function of the specimen type. Poisson’s ratio was determined as the angle of the slope of the transverse strain curve as a function of the longitudinal strain. Calculations were performed for each specimen tested using Excel. The value given in Table 3 is the average obtained after rejecting extreme values.

The obtained values of failure stresses (*σ_f_*) (*σ_f_*) were averaged for each test and presented in the form of diagrams (Figure 9) with standard deviation bars. The same method was used to visualize the longitudinal failure strains (*ε_f_*) (*ε_f_*) and their standard deviation, which are presented in Figure 10.

An example of a damaged specimen is shown in Figure 11a. A set of damaged specimens in the uniaxial tensile test specimens made of E601 epoxy resin with the addition of ESO3 is shown in Figure 11b. On the other hand, Figure 11c shows a set of damaged specimens made of E601 with the addition of MC10.

To determine the optimal content of the organic additive, the average values of Young’s modulus, Poisson’s ratio, failure stress and failure strain were determined for the E601 resin as reference values (E_ref_, ν_ref_, σ_fref_ and ε_fref_). The average values of the same quantities obtained for specimens with additives were related to these values. The results in the form of ratios are shown as graphs in Figure 12.

## 5. Conclusions

We do not recommend ethyl cellulose (EC) as an additive to the epoxy resin due to its poor miscibility. The best miscibility with epoxy resin and the best-looking test specimens were observed when using carboxymethyl cellulose (CMC) additives.Both maximum values of strains and stresses in the stress–strain curves, higher than for epoxy resin, were observed for epoxy resins modified with epoxidized soya oil. The highest stresses were detected for epoxy E601 resin modified with 3 phr soya oil (ESO3), whereas the highest strains occurred for epoxy modified with 10 phr soya oil (ESO10) specimens.The smallest standard deviation values of Young’s modulus, Poisson’s ratio and stress and strain values were observed for epoxy resins modified with 3 phr added epoxidized soya oil (ESO3).Despite the imperfections of the prepared specimens and the small population of the tests carried out, the obtained results seem interesting and indicate the desirability of further extended research in biodegradation testing. We propose continuing research with epoxy resin modified with 3 phr epoxidized soya oil (ESO3) and 10 phr of methyl cellulose (MC10).

## Figures and Tables

**Figure 1 materials-16-01854-f001:**
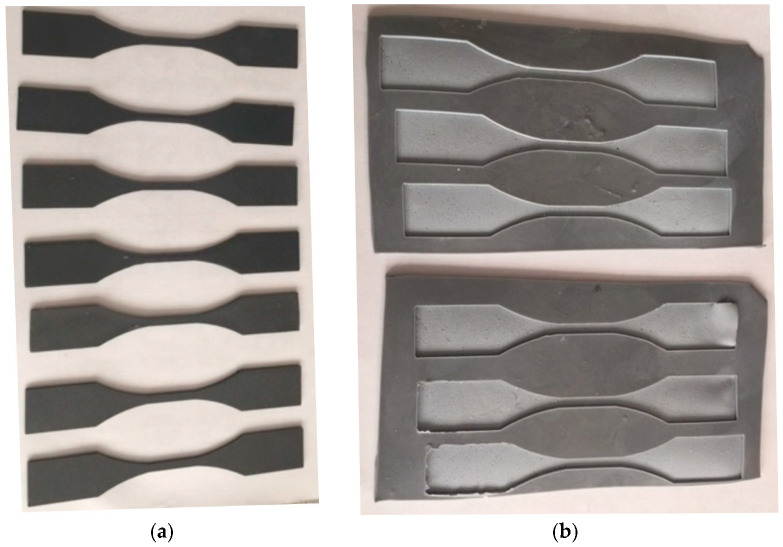
(**a**) Test reference specimens made by 3D printing and (**b**) silicone molds made from Polastosil.

**Figure 2 materials-16-01854-f002:**
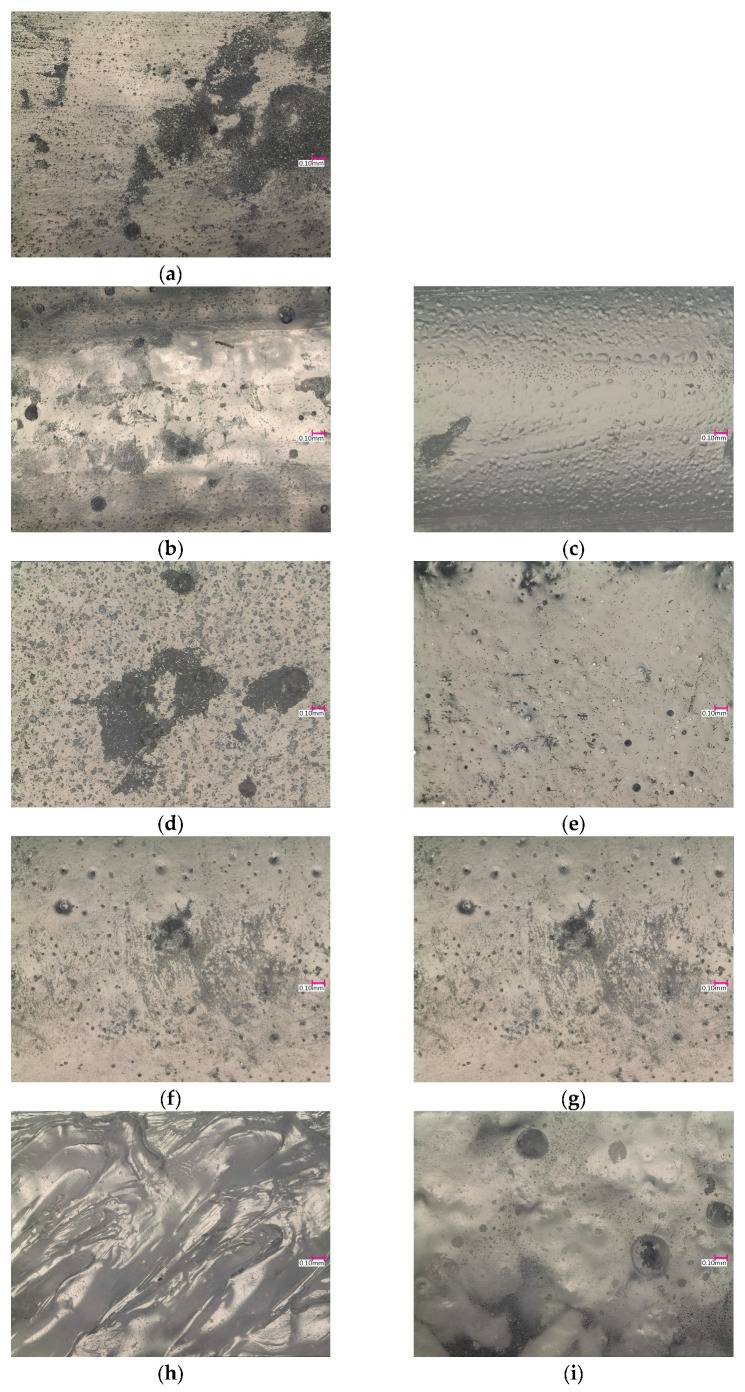
Microscopic photos of the fragments of specimens made with mixtures of (**a**) epoxy resin E601 and additive, (**b**) ESO3, (**c**) ESO10, (**d**) MC3, (**e**) MC10, (**f**) CMC3, (**g**) CMC10, (**h**) EC3 and (**i**) EC10.

**Figure 3 materials-16-01854-f003:**
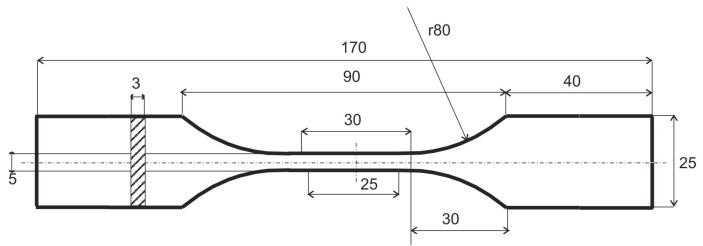
Nominal dimensions of the test specimen in mm.

**Figure 4 materials-16-01854-f004:**
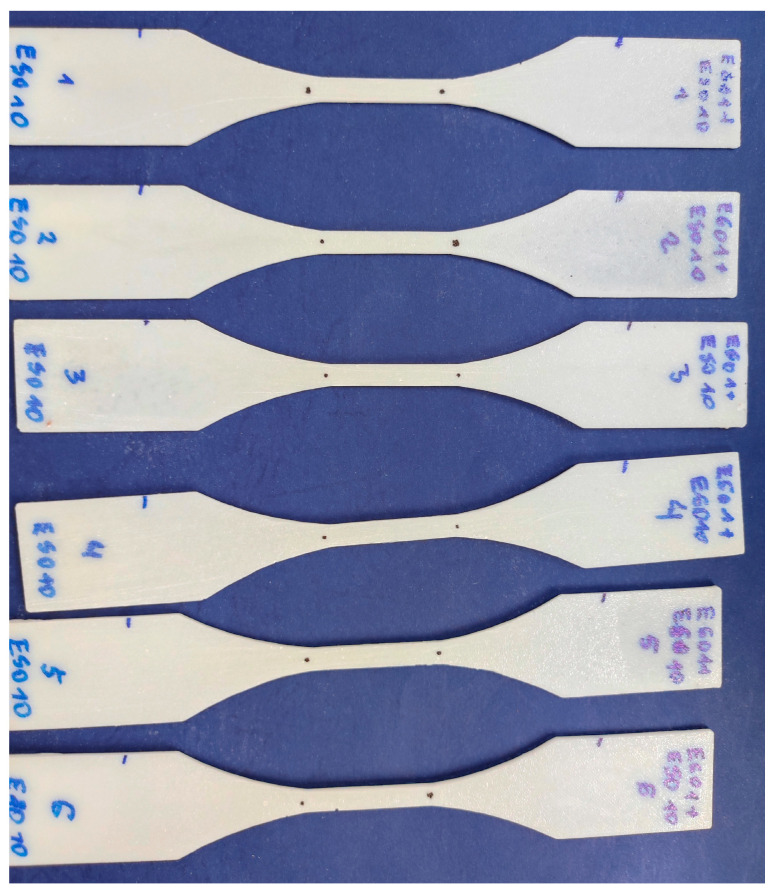
Exemplary set of test specimens prepared for testing that are made of epoxy resin E601 with the addition of ESO10.

**Figure 5 materials-16-01854-f005:**
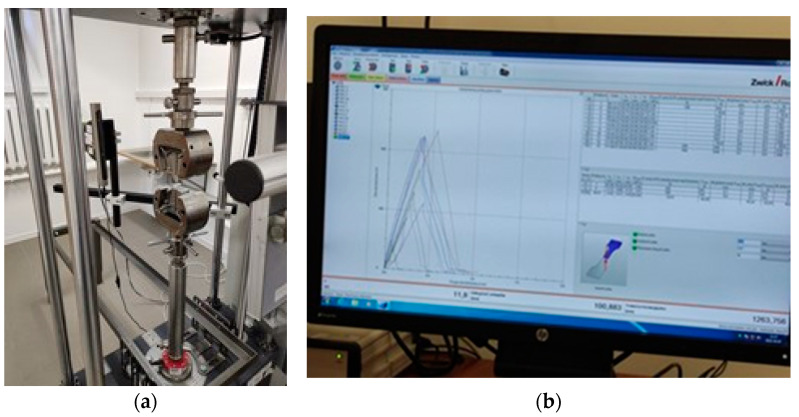
(**a**) KAPPA 50 DS electromechanical loading system with a ZwickRoell video extensometer and (**b**) specialized software testXpert ZwickRoell.

**Figure 6 materials-16-01854-f006:**
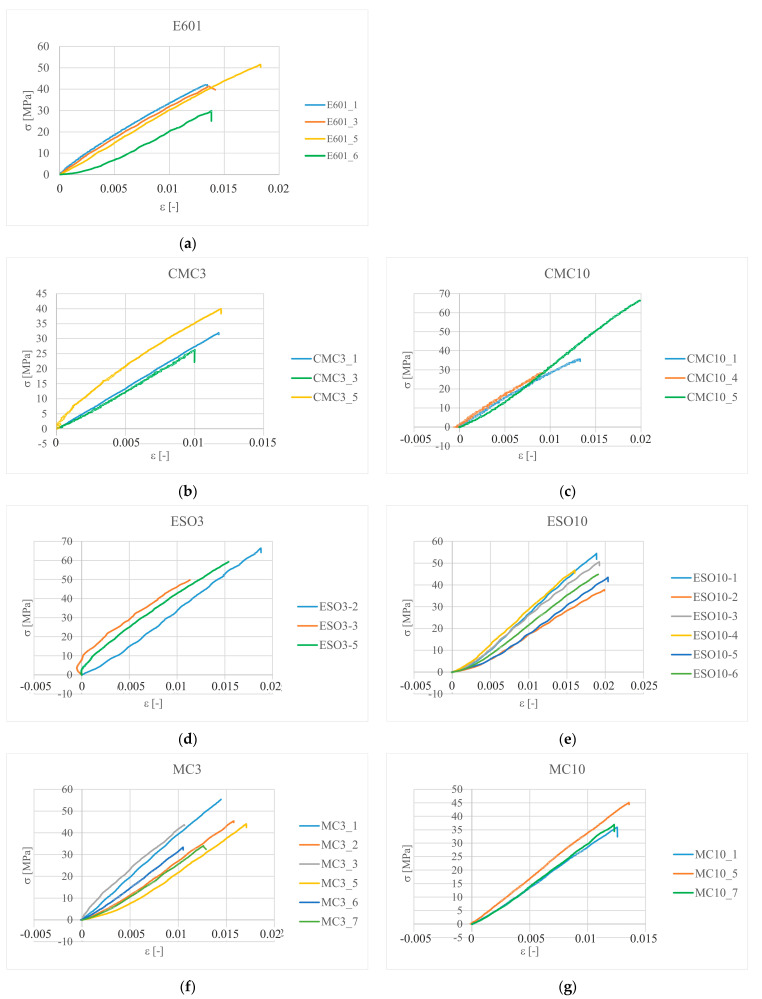
Sets of tensile stress–strain curve for (**a**) epoxy resin without fillers, (**b**) CMC3, (**c**) CMC10, (**d**) ESO3, (**e**) ESO10, (**f**) MC3, (**g**) MC10, (**h**) EC3 and (**i**) EC10.

**Figure 7 materials-16-01854-f007:**
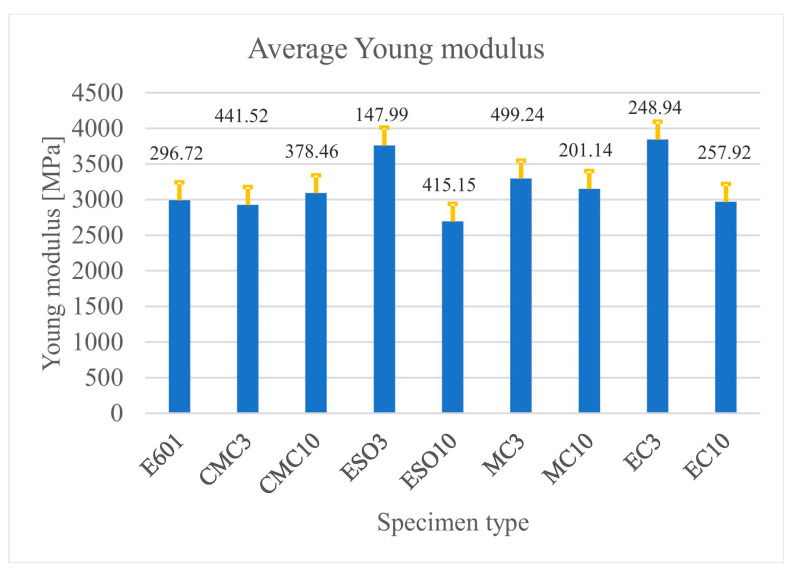
Average values of Young’s modulus with standard deviation bars as a function of the specimen type.

**Figure 8 materials-16-01854-f008:**
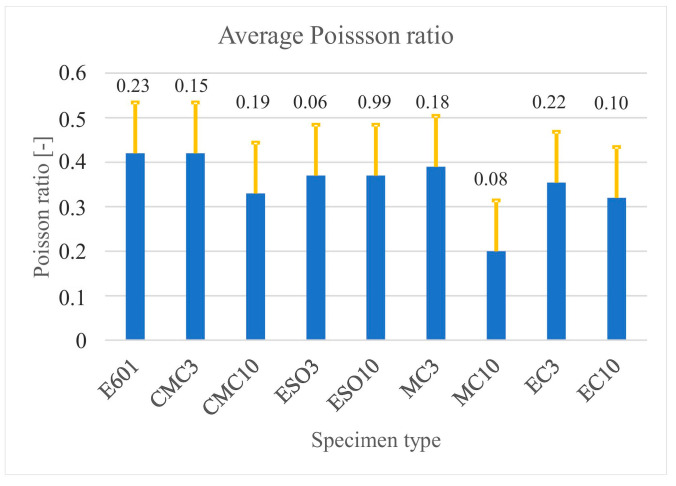
Average values of Poisson’s ratio with standard deviation bars as a function of the specimen type.

**Figure 9 materials-16-01854-f009:**
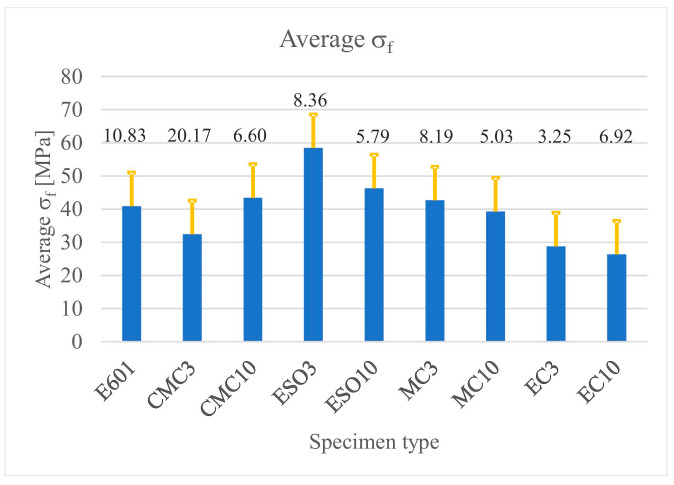
Average values of failure stress with standard deviation bars as a function of the specimen type.

**Figure 10 materials-16-01854-f010:**
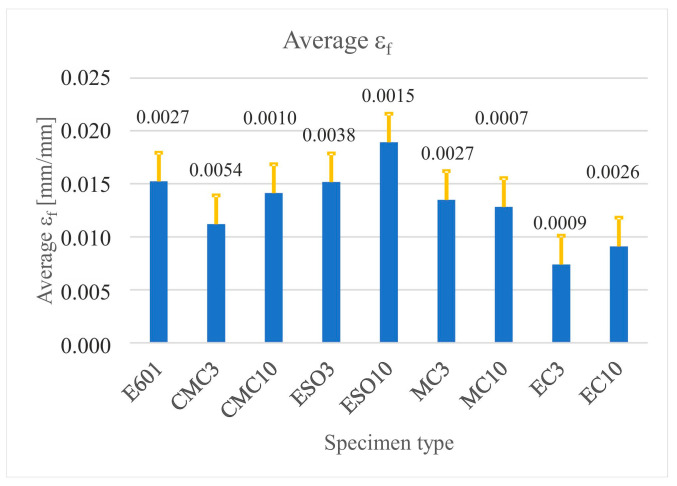
Average values of failure strain with standard deviation bars as a function of the specimen type.

**Figure 11 materials-16-01854-f011:**
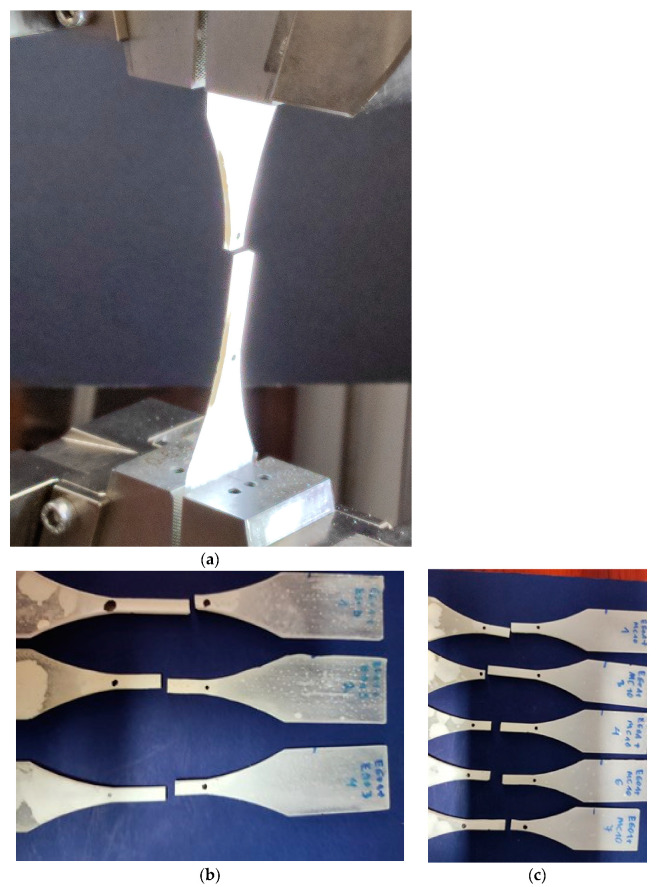
(**a**) An example of a damaged specimen. Set of damaged specimens in the uniaxial tensile test specimens made of (**b**) E601 + ESO3 and (**c**) E601 + MC10.

**Figure 12 materials-16-01854-f012:**
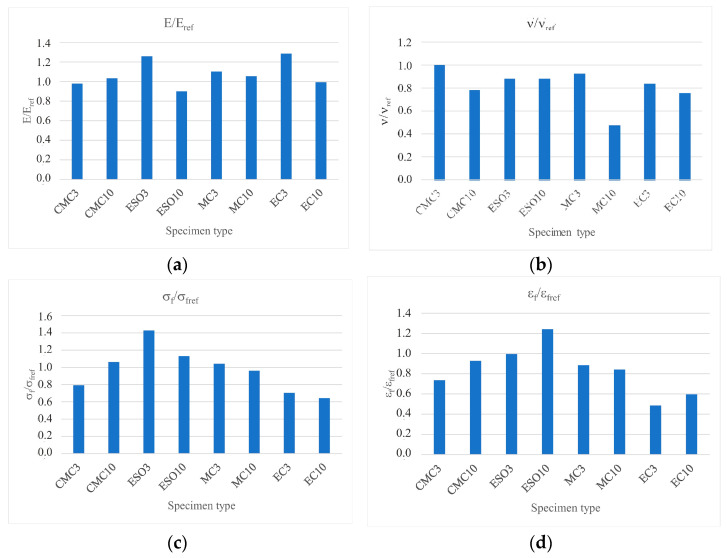
Ratios of (**a**) E/E_ref_, (**b**) ν/ν_ref_, (**c**) σ_f_/σ_fref_ and (**d**) ε_f_/ε_fref_.

**Table 1 materials-16-01854-t001:** Composition of epoxy/additive mixtures for determining mechanical properties.

No.	Epoxy Resin	Additive	Phr	Remarks
1	Epidian 601	Without additive	-	-
2	Epidian 601	ESO	3	Ref. [28]
3	Epidian 601	ESO	10	Ref. [28]
4	Epidian 601	MC	3	Ref. [20]
5	Epidian 601	MC	10	Ref. [20]
6	Epidian 601	CMC	3	Ref. [20]
7	Epidian 601	CMC	10	Ref. [20]
8	Epidian 601	EC	3	Ref. [20]
9	Epidian 601	EC	10	Ref. [20]

**Table 2 materials-16-01854-t002:** Basic properties of Epidian 601 resin (low viscosity liquid) [29].

Property	Unites	Value	Remarks
Colour	-	Yellow, slightly opaque	-
Boiling point	°C	210	Starts decomposition
Fire point	°C	180	-
Epoxide number	Mol/100 g	0.5–0.55	-
Density	g/cm^3^ at 25 °C	1.14	-
Viscosity	mPas at 25 °C	700–1100	-
Gelling time	min	40	After addition of 13 phr Z1; RT

**Table 3 materials-16-01854-t003:** Average values of Young’s modulus, Poisson’s ratio, failure stresses and strains, and the number of specimens.

Name	E (MPa)	SD E	*ν* (-)	SD *ν*	*σ_f_* (MPa)	SD *σ_f_*	*ε_f_* (MPa)	SD *ε_f_*	No. of Specimens
E601	2991.5	296.72	0.42	0.2281	40.94	10.83	0.0152	0.0027	4
CMC3	2926.4	441.52	0.42	0.1457	32.49	20.17	0.0112	0.0054	3
CMC10	3091.7	378.46	0.33	0.1909	43.48	6.60	0.0141	0.0010	3
ESO3	3761.3	147.99	0.37	0.0635	58.50	8.36	0.0152	0.0038	3
ESO10	2692.6	415.15	0.37	0.0988	46.32	5.79	0.0189	0.0015	6
MC3	3297.3	499.24	0.39	0.1832	42.69	8.19	0.0135	0.0027	6
MC10	3152.1	201.14	0.2	0.0794	39.33	5.03	0.0128	0.0007	3
EC3	3845.5	284.94	0.35	0.22	28.81	3.25	0.0074	0.0009	5
EC10	2969.6	257.92	0.82	0.26	26.37	6.92	0.0091	0.0026	4

## Data Availability

Not applicable.

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
