# Peer review of "Mechanical Testing of Epoxy Resin Modified with Eco-Additives"

_materials, 2023, doi:10.3390/ma16051854_

Round 1

Reviewer 1 Report

The writing of this paper is like a test report than a scientific paper. The abstract and conclusion must be rewritten, and the results and analysis are not sufficient. Overall, the writing quality is poor, which cannot provide readers with some meaningful results. Therefore, it is not recommended to publish it.

Author Response

Answer to Reviewer 1

Thank you for opinion. Yes, paper in this form is more like a report, not a scientific article. We have changed abstract, conclusions and discussion. English form was also verified by a native speaker.

Reviewer 2 Report

In this contribution, the authors investigated the tensile performance of epoxy resins with various additives. Although this topic is inspiring to the readership of Materials, this research overlooked critical details. The following questions and comments need to be addressed before further decisions.

1. The bio-degradability of epoxy resin is emphasized in both the title and the Introduction. However, the bio-degradability of epoxy resin and additives needs either literature support or experimental data.

2. How are the additives mixed with epoxy? Why is introducing EC not successful?

3. Why do Figure 2 and 3ab show roughness on the surface? Does the roughness impact the consistency of tensile results? Why does ESO3 show a rough surface, But the smallest standard deviation in Young's modulus and failure stress?

4. Figure 1b and Line 82 introduce 3D printed reference specimens made from PET. What are the PET reference specimens for?

5. The numbers of specimens, 3 or 4, are lean for reliable standard deviations.

6. Other amendments:

In Table 2 and 3, the units of strains and their standard deviations should be %, not MPa.

In Figure 10b, the title should be the standard deviation of failure strains, instead of average failure strains.

Author Response

Thank you for opinion. Yes, paper in this form is more like a report, not a scientific article. We have changed abstract, conclusions and discussion. List of references was enriched. Some corrections to the Experimental part were done according to the Reviewer’s 2 suggestions (ISO/ASTM standards were added; Table 2 was merged with Table 3; error bars were added in Figures 7 and 8). English form was also verified by a native speaker.

Reviewer 3 Report

  • In general, the English level must be updated, as now of poor quality which affects readability of the manuscript
  • It is advised to extend the abstract, as now very limited.
  • The introduction is vague and does not fully focus on the nature of the manuscript (biodegradable resins). This must be improved
  • Line 30 “ The matrix ensuring the integrity of the material is the most influencing mechanical 30 strength of composites.” Is a strage statement as this seems that only matrix is important. Clarification is needed.
  • Line 47 “For years, researchers from India …”. And what about other researchers outside India?
  • In research methods, it only focusses on tensile strength. Is this the only important aspect as for composites, as in general mainly aspects such as impact and crack propagation are important to evaluate.
  • It is mentioned that a silicone mold has been applied, whereas the original sample is 3D printed. In scope of tensile evaluation, this cannot be advised as 3Dpriting samples will create micro-errors later on resulting in micro crack initiation. Why no standard samples have been used as pre-product?
  • Within experimental methods, equipment is not well defined.
  • Line 124, determination of modulus. Why not use of extensometer? This is unclear
  • Table 2, why for each mixture a difference in samples have been investigated?
  • Figure 7, why not combining both modulus/poison ratio and standard deviation within one graph. This is confusion to evaluate the outcome of experimental testing. Larger variations are also not well explained. This can be caused due to the type of sample preparation and must be clarified.
  • It is strongly advices to optimize the “Conclusions” as now no detailed evaluation of the results in correlation with the selected materials and methods.

Author Response

Thank you for opinion. We have changed abstract, conclusions and discussion. New title of the paper was also proposed. This was one year project as initial stage of bigger one's. Further studies are planned, including numerical analyses.

Only references specimens used to prepare silicone mold were 3D printed. Original specimens were made from mixtures. Contrary to expectations, at the applied stretching speed of 2 mm/min, the breakage of the samples occurred as expected. A few photos specimens after failure were attached to paper.

A machine with a video extensometer and specialized software was used for the tests, enabling the simultaneous measurement of the elongation and change in the width of the specimen in a given area. A more detailed description of the equipment used and the method of conducting the research was included in the article.

It is possible that due to the low level of the English language, some phrases have been misunderstood. Therefore English form was also verified by a native speaker.

Reviewer 4 Report

The manuscript entitled "Development of biodegradable epoxy resin and determination of its tensile strength" reports the mechanical characterization of several samples of epoxy resins containing different additives which could improve the biodegrabability of the matrix.

In my view, the topic of the submitted work could be of some interest but just the results of the tensile characterization are reported in the manuscript. Furthermore, the comments about the observed behaviors are poorly developed and the manuscript looks like a scientific report. Therefore, in my opinion, the manuscript is not publisheable in the current state.

Some other issues:

- the Abstract is too short and poor of information. Please add the main results of the work and some conclusions.

- the list of references needs to be enriched.

- in the experimental part, the Authors reported "The photos of the middle fragment of the created specimens made of mixtures of epoxy resin and additives". It should be better explained the meaning of "middle fragment".

- in the experimental part, the Authors shoud specify the conditions used to perform the tensile characterization and the ISO/ASTM standard followed.

- to improve the readability of the data reported in Table 2, the standard deviation data should be reported in the same Table. Therefore, I suggest to merge Table 2 and 3.

- Similarly, data reported in Figures 7 and 8 should be implemented with error bars, showing the standard devation values.

Author Response

Thank you for detailed opinion. We have changed title of the paper, we have added interesting paper of Webster; we have corrected error in Table 3; more details about experimental part (mixing procedure) were also added. abstract, we have corrected Figure 7 and 8 and we have tried to explain poor results obtained with cellulose additives. Remark number 4 – tests for biodegradation – yes, we agree this is crucial in that project but we want to run these experiments and report them in a next paper. English form was also verified by a native speaker.

Reviewer 5 Report

Manuscript Number: Materials-2096876

The manuscript written by Derewonka et al. titled “Development of biodegradable epoxy resin and determination of its tensile strength” studied the effect of biobased/biodegradable epoxy additives on the mechanical properties of commercial epoxy resin. I want to suggest the following comments and clarify a few concerns which will further improve the article and make it suitable for Materials.

1.      The title of the paper overestimated the work reported by the authors. The use of biobased epoxy oils is previously reported by other authors, therefore, the “development of biodegradable epoxy resin” part of the title needs to be removed and the title should be rewritten.

2.      The abstract could be more informative, and include some trends that were observed in the tensile testing.

3.      Thermal and mechanical properties of biobased epoxy resin in comparison to DGEBA resin were studied by Webster et al. (https://doi.org/10.1016/j.polymer.2021.124191). This reference could be suitable to add to the introduction.

4.      As the authors have not carried out biodegradation testing, if such testing has been previously reported for the tested formulations, please report the results in the introduction or in Table 1.

5.      As both the mechanical and thermal properties (softening point, Tg) of epoxy composite depend on the nature of additives, in that case, any reason the authors only focused only on the mechanical properties and not studied the thermal properties. If  possible please include the thermomechanical properties using DMA/DSC of the formulations to understand the effect of flexible epoxy oils on the overall network.

6.      In table 3. 8/36 should be 8.36/

7.      Please add how the mixing of epoxy oil was carried out in the DGEBA-based commercial resin.

8.       As I understood to cure the resin TEPA was used, what was the EEW to AHEW ratio?

9. Figures 7-10 reflect the same data previously reported in table 3. It can be removed, or if the authors want to keep them then the resolution needs to be improved.

10.  The authors have made the conclusion but any reason why ESO works the best than the cellulose additives. Please add.

Author Response

Thank you for opinion. Yes, paper in this form is more like a report, not a scientific article. We have changed abstract, conclusions and discussion. New title of the paper was also proposed.

Thank you for interesting paper. It will be included to references. List of references was enriched. This was one year project as initial stage of bigger one's. Further studies are planned, including numerical analyses.

The results from the tensile tests of specimens made of E601+EC3 and E601+EC10 mixtures have been included in the graphs. I hope it will be interesting.

Round 2

Reviewer 1 Report

The authors have made the improvements to the paper compared to the original version, but the following comments should be considered to make necessary supplements and explanations.

1.     Abstract, the effect mechanism of fillers on epoxy resin should be further clarified. In addition, the content of fillers and the quantitative effect on mechanical properties should be further supplemented.

2.     Introduction, for the selection of fillers, the authors should consider the advantages and disadvantages of organic fillers and inorganic fillers. For inorganic filler, it can significantly improve the mechanical properties and long-term properties of epoxy resin, but it may not be degradable. In contrast, organic fillers can be degraded. However, its effect on improving the mechanical properties of epoxy resin may not be as good as that of inorganic fillers. Further comparison and analysis of the action mechanism between the two fillers and epoxy resin, the content of fillers and the existing form are recommended. Please review the relevant research below to make necessary supplements. Nanomaterials, 2021, 11, 1234. Materials Today Communications, 2020, 24: 101360. Polymers. 2022, 14, 1087.

3.     In the last paragraph of the introduction, the authors should briefly introduce the current main research work, as well as the main contributions and innovations.

4.     The fourth part should be the results and discussion.

5.     Why is the tensile stress-strain curves of epoxy resin with and without the fillers not provided? These curves are important to analyze the mechanical properties and the failure characteristics of materials.

6.     The clarity of the picture should be improved.

7.     For different fillers, the mechanical properties of the composites are different. So how to determine the optimal content, type of fillers and the action mechanism between fillers and epoxy resin? Please explain this.

8.     The conclusion should be reduced, only containing 3-4 points.

Author Response

Thank you for your valuable comments and guidance.

  1. The abstract has been expanded.
  2. The introduction of inorganic fillers significantly improves the stability of the epoxy's mechanical and thermal properties and water absorption. This leads to an extension of the life of the structure. However, the goal of this work is to use renewable, organic additives that will result in materials and structures that reach their service limits during operation being recovered and reused in a closed cycle.
  3. The Intoduction also adds a paragraph giving some of the most current literature in the field of biodegradable epoxies.
  4. The fourth part is the results and discussion.
  5. The tensile stress-strain curves of epoxy resin with and without the fillers were added to section 3.
  6. Clarity of pictures were improved.
  7. To determine the optimal content of the organic additive, the average values of Young's modulus, Poisson's ratio, failure stress as well as failure strain were determined for E601 resin as reference values (Eref, nref, sfref, efref). To these values were related the average values of the same quantities obtained for specimens with additives. The results, in the form of ratios, are shown as graphs in Figure 13.

8. The conclusions were reduced to 4 points.

Reviewer 2 Report

My questions in the first revision have been addressed. Additionally, the following comments are expected to assist in further improving this manuscript.

1. Figures on the same topic are expected to be combined, such as Figure 2 and 3, Figure 7 and 12, etc.

2. The uniaxial tensile test is the core evaluation of all specimens; therefore, examples of stress-strain curves are expected to be presented. Specifically, the curves in Figure 6b are essential to show the mechanical performance.

3. In Table 3, why is the Poisson’s ratio of EC10, 0.82, significantly greater than that of other specimens?

4. In Figure 8, 9, 10, and 11, the results should be in a form of columns and error bars standing for the average ± standard deviation, instead of stacking the standard deviation on average.

Author Response

Thank you for your valuable comments and guidance.

  1. Figures were combined.
  2. The tensile stress-strain curves of epoxy resin with and without the fillers were added to section
  3. The high value of the Poisson's ratio for the mixture of E601 with EC10 is probably due to the methodology of its determination and the number of specimens tested. The Poisson's ratio was determined as the angle of slope of the transverse strain curve as a function of longitudinal strain. Calculations were performed for each specimen tested using Excel. The value given in Table 3 is the average obtained after rejecting extreme values.
  4. Figures 8, 9, 10, and 11 were corrected.

Reviewer 4 Report

Despite the intensive revision made by the Authors, I'm still convinced that the manuscript does not deserve publication on Materials, mainly due to the fact that the manuscript reports solely the mechanical characterization of samples based on commercial formulations. 

Author Response

Special thanks for detailed opinion about our paper. We agree that some work has to be added, about biodegradability, but we wanted to concentrate on two topics. First – in numerous small companies people are anxious to read about mechanical properties of epoxy composites with added biodegradable additives. Secondly we wanted to concentrate on easy available components. We plan in the near future to write a second paper including all important suggestions.

Reviewer 5 Report

Thank you for improving the manuscript. I have a few minor corrections that I would like to provide.

_Please make sure that Figure 1. (a) Test reference specimens made by 3D printing, (b) silicone molds made from Polastosil, is not interchanged, looks like (a) is the mold and (b) are the specimens.

_In line 121 "The effect of organic additives 121 on the structure of the modified epoxy resin is shown in Figure 3." What effect the authors are referring to please elaborate.

_Please add the scales for the SEM pics in FIgure 3.

_Please improve the quality of the figures in section 4. Also, use the same fonts as the main body text.

_Instead of the photo of the monitor showing the graphs in 6(b), please provide the Figure with the XY plots.

_Please offer a material and methods section which will be helpful for the readers to repeat the experiments.

Author Response

Thank you for your valuable comments and guidance.

  1. There have to be holes in the mold that match the shape and dimensions of the specimens to be created. The reference specimens shown in Figure 1a were used to create them.

2 and 3. The scale on each picture allows to determine the size and distribution of additives. Inserting a scale is more practical than a scale in pics.

  1. The quality of the figures in section 4 were improved.
  2. The tensile stress-strain curves of epoxy resin with and without the fillers were added to section 3.
  3. Materials and methods are described in section 2.

Round 3

Reviewer 1 Report

The authors have almost all replied to the comments and recommended to accept the current paper.

Author Response

Thank you for your recommendation.

Reviewer 2 Report

This manuscript has been dramatically improved from the second round of revision. The efforts are appreciated. However, the following comments are expected to be addressed.

1. The error bars in Figure 8, 9, 10, and 11 are not set correctly.

2. Again, in Table 3, the Poisson's ratio of EC10 as 0.82 is too high, and needs to be verified. The typical Poisson's ratio of polymeric materials is 0.3-0.4. Unless why EC10 is special is reasonably addressed, a Poisson's ratio of 0.82 is not acceptable. 

Author Response

Thank you for your guidance and opinion.

  1. The error bars in Figure 8, 9, 10, and 11 were corrected.
  2. The Poisson's ratio of EC10 was corrected to 0.32. It was mistake in my calculations. Thank you for bringing it to my attention again.

Reviewer 4 Report

The paper can be accepted in the current form

Author Response

Thank you for your recommendation.
